# Recognition of Protein Pupylation Sites by Adopting Resampling Approach

**DOI:** 10.3390/molecules23123097

**Published:** 2018-11-27

**Authors:** Tao Li, Yan Chen, Taoying Li, Cangzhi Jia

**Affiliations:** 1School of Transportation Management, Dalian Maritime University, Dalian 116026, China; litao@wti.ac.cn (T.L.); chenyan@dlmu.edu.cn (Y.C.); 2China Waterborne Transport Research Institute, Beijing 100088, China; 3College of Science, Dalian Maritime University, Dalian 116026, China; cangzhijia@dlmu.edu.cn

**Keywords:** fuzzy undersampling, machine learning, principal component analysis, protein pupylation, sequence information

## Abstract

With the in-depth study of posttranslational modification sites, protein ubiquitination has become the key problem to study the molecular mechanism of posttranslational modification. Pupylation is a widely used process in which a prokaryotic ubiquitin-like protein (Pup) is attached to a substrate through a series of biochemical reactions. However, the experimental methods of identifying pupylation sites is often time-consuming and laborious. This study aims to propose an improved approach for predicting pupylation sites. Firstly, the Pearson correlation coefficient was used to reflect the correlation among different amino acid pairs calculated by the frequency of each amino acid. Then according to a descending ranked order, the multiple types of features were filtered separately by values of Pearson correlation coefficient. Thirdly, to get a qualified balanced dataset, the K-means principal component analysis (KPCA) oversampling technique was employed to synthesize new positive samples and Fuzzy undersampling method was employed to reduce the number of negative samples. Finally, the performance of our method was verified by means of jackknife and a 10-fold cross-validation test. The average results of 10-fold cross-validation showed that the sensitivity (Sn) was 90.53%, specificity (Sp) was 99.8%, accuracy (Acc) was 95.09%, and Matthews Correlation Coefficient (MCC) was 0.91. Moreover, an independent test dataset was used to further measure its performance, and the prediction results achieved the Acc of 83.75%, MCC of 0.49, which was superior to previous predictors. The better performance and stability of our proposed method showed it is an effective way to predict pupylation sites.

## 1. Introduction

Pupylation is a prokaryotic analog of ubiquitination whose prokaryotic ubiquitin-like protein (Pup) separates intracellular proteins under the action of enzymes and specifically modifies the target protein [1,2]. Its proteasome-independent functions help in the regulation of DNA repair mechanisms, particularly recent advances on its regulatory role for careful series of functions that gives rise to a therapeutic solution in cancer chemotherapy [3,4,5]. Pup is an identified posttranslational small modifier in prokaryotes [1,2] that usually attaches to the substrate lysine via isopeptide bonds, and this process is called pupylation. Although the functions of pupylation and ubiquitylation are similar, the enzymologies involved in these processes are different [1,2,6]. Since experimental methods are laborious, it is essential to improve the current computational methodologies to provide direction for further research on intriguing research questions. An example of this is the works on understanding the stability of ERCC1 DNA repair protein, a biomarker of several advanced cancer diseases. This protein functions in multiple DNA repair pathways and certain mutations in this protein and its partner had drastic consequences for the protein complex stability [3,4,5,6,7,8,9]. Functional and structural studies have determined the significance of the complex integrity and ubiquitination/deubiquitination events for controlling the function of the protein during DNA damage response [1,2,6,7,8]. These findings persuade new therapeutic solution in cancer chemotherapy. Thus, it is important to devise accurate computational methodologies for scientists to ultimately predict the probability of different regulatory pathways that control protein function and disease severities [6,8,9]. Several proteomics methods have been proposed for identifying pupylated proteins and pupylation sites [10,11,12,13,14,15,16,17]. However, pupylation sites of many experimentally identified pupylated proteins are still unknown. Accurate identification of pupylation sites is an essential first step to better understand the mechanism underlying protein pupylation. A number of large-scale proteomics techniques have been used for predicting pupylation sites, which are time-consuming and laborious. Therefore, a series of other effective and accurate prediction methods have been proposed for predicting pupylation sites. 

The Group-based Prediction System was used by Lin to design the predictor of GPS-PUP for predicting the pupylation sites [11]. The predictor iPup [12] was designed by combining the composition of k-spaced amino acid pair (CKSAAP) feature and support vector machines (SVMs). Zhao et al. [13] created a predictive model with five features and also adopted feature selection methods to find the optimal feature set. Chen et al. [14] proposed a model PupPred, which was also an SVM-based predictor. It used amino acid pair features to encode lysine central peptides and combined a series of features to improve its predictive performance. Hasan et al. [15] constructed a predictor called pbPUP that used the profile-based composition of k-spaced amino acid pairs (pbCKSAAP) to represent the sequence information around the pupylation site. Jiang et al. [16] created a predictor called PUL-PUP that combined the positive-unlabeled learning technique with CKSAAP to predict pupylation sites. A structured and searchable database PupDB was used for integrating information of pupylated proteins and sites, protein structures, and functional annotations for the management and analysis of pupylation sites due to a large number of newly identified pupylated proteins and sites [16]. Recently, Nan et al. [17] proposed the enhanced positive-unlabeled learning algorithm for predicting pupylation sites.

In this study, the Pearson correlation coefficient was used to evaluate the relevance among amino acid pairs based on amino acid composition information. Then, according to the descending order, the amino acid pair features were added by a step of 20 to get the best prediction results by means of jackknife test. Then the selected 320 amino acid pairs were combined with TOP-n-gram [18], adapted normal distribution bi-profile Bayes (ANBPB) [19], and parallel correlation pseudo amino acid composition (PC-PseAAC) [20] to construct a multiple feature vector for the query protein peptide sequence. Because of the imbalance between the number of positive and negative samples (183:2258), the K-means principal component analysis (KPCA) oversampling technique, firstly proposed by Jia and Zuo [21], was applied on the positive training dataset for oversampling, and the synthetic samples were added to the original positive training dataset as a new positive training dataset. The fuzzy undersampling (FUS) method [21,22] was applied to the negative dataset to reduce noise negative samples, and the selected negative samples were used as a new negative training dataset. At last, the 377 pupylated sites (positive samples) and 365 non-pupylated lysine sites (negative samples) were used to train and test our model. Moreover, the performance of our method was verified by means of jackknife, 10-fold cross-validation test and an independent dataset. When compared with other existing predictors, all of the results showed better performance and stability, proving as an effective way to predict pupylation sites.

## 2. Results and Discussion

### 2.1. Pearson Correlation Coefficient for Feature Selection

The Pearson Correlation Coefficient [23,24] is a measurement for the linear relationship among distance variables. If a positive linear correlation exists between two variables, their metric approaches 1. The Pearson Correlation Coefficient was used to find out the most closely related amino acids. First, extracting amino acid composition (AAC) was used for feature extraction on the training dataset. Second, the Pearson Correlation Coefficient was employed to calculate the correlation coefficient among amino acids based on AAC values, and its values were sorted in a descending order. Finally, the amino acid pair composition (AAPC) was selected according to the order. The value of Pearson Correlation Coefficient was 1 for the amino acid pair AA, CC, ..., YY, and so these 20 amino acid pairs of the same amino acid were reserved to combine with others. The prediction on the jackknife test on the combination of 20, 40, ..., 380, and 400 features was performed by a step of 20, and the detailed results are shown in Appendix A. The best results were obtained when the first AAPC (320) features were selected; sensitivity (Sn) was 81.42%, specificity (Sp) was 76.44%, accuracy (Acc) was 78.10%, and Matthews Correlation Coefficient (MCC) was 0.55. However, the prediction results were not monotonous; this might be due to the same parameter used for different dimension features. 

### 2.2. Combination of KPCA and FUS for Training Set Balancing

As described in Section 2.1, the training dataset comprised 183 positive samples and 2258 negative samples which is the same as [15,17] to guarantee the contrast of the experiment.

The ratio of positive and negative samples was 1:12, leading to biased results and inaccurate data. The KPCA and fuzzy undersampling (FUS) oversampling and undersampling methods were first proposed by Jia and Zuo [21]. They were effective in identifying protein O-GlcNAcylation sites. Therefore, in this study, KPCA and FUS were also employed to solve the imbalance between positive and negative training datasets. First, FUS was used to remove the redundant negative samples, and the remaining 365 nonpupylation protein peptides were used as the new negative training dataset. Then, KPCA was applied on the positive samples to divide them into three clusters (k = 3). The experiment was repeated several times. The results of clustering were added to the original positive samples so that the ratio of the positive samples and the negative samples was approximately 1:1. The following results were obtained based on KPCA and FUS. 

The performance illustrated that the results obtained from the original unbalanced training dataset were biased toward the larger number of classes. For example, on the 320-dimensional selected feature vector, Sn was 0 and Sp was 100%; it was the worst performance for the prediction model. However, after using KPCA and FUS on the training dataset, Sn was 81.42% and Sp was 76.44% shown in Table 1, indicating that the unbalanced data had a great influence on the experimental results.

### 2.3. Predictive Performance Improvement

A variety of feature extraction methods can be found on the Web server (http://bioinformatics.hitsz.edu.cn/Pse-in-One2.0/) [25]. After several trials, TOP-n-gram, ANBPB, and PC-PseAAC feature extraction methods were used to combine with 320-dimensional features extracted by AAPC, and the results are shown in Table 2.

The Pearson correlation coefficient was used to filter out the AAPC (320) dimension features from the 400-dimensional features so as to remove the redundant information. Although the results were improved, the value of MCC was slightly lower. AAPC (320) was combined with other features to increase the value of MCC. When AAPC (320) was combined with TOP-n-gram and ANBPB, Sn, Acc, and MCC improved greatly. Especially when AAPC (320) was combined with TOP-n-gram, MCC increased by 0.3037, but Sn slightly reduced. When AAPC (320) was combined with PC-PseAAC, Sn increased by 2.73%. Combining AAPC (320) with TOP-n-gram and PC-PseAAC, Sn increased by 6.01%. After several experiments, AAPC (320) was combined with TOP-n-gram, ANBPB, and PC-PseAAC, with the MCC of 0.96, which was higher by 0.41 than the previous value of AAPC (320); the accuracy was 98.18%. Furthermore, Sn and Sp increased by 13.12% and 23.56% compared with AAPC (320). The results indicated that the predictor achieved better prediction performance.

### 2.4. Comparison between the Proposed Method and Other Prediction Methods

The proposed method was compared with other methods by tenfold cross-validation, including EPuL [17], PUL-PUP [16], PSoL [26], and SVM balance [15] on the training dataset to evaluate the effectiveness of the proposed method for pupylation site prediction. Table 3 presents the comparison among EPuL, PUL-PUP, PSoL, and SVM balance.

The SVM balance method can solve the imbalance of datasets, but its negative samples obtained from unannotated lysine sites are inaccurate. PUL-PUP and PSoL use unreliable negative datasets, whose features are not optimal. EPuL uses a series of options to obtain the reliably negative datasets; however, its accuracy is low because it does not integrate other features and the imbalance dataset is selected randomly. Figure 1 shows that the proposed method achieved a higher accuracy compared with EPuL for running 10 times of tenfold cross-validation results, with Sn of 6.32% and Sp of 4.35%. Furthermore, the Acc and MCC of the proposed method were 95.09% and 0.9063, respectively, which were 4.85% and 0.0963 higher than those of EPuL. For the EPuL web-server is unavailable, we listed the detailed 10-fold cross-validation results of our method, PUL-PUP, PSoL, and SVM in Appendix A.

### 2.5. Performance on the Independent Test Dataset

In this study, the proposed method achieved a better performance, indicating that the Pearson correlation coefficient, KPAC, and FUS were effective in predicting the pupylation sites. An independent test dataset containing 20 proteins, which included 29 experimentally validated pupylation sites and 408 nonannotated pupylation sites, was selected to further verify the effectiveness of the proposed method. Although the positive and negative datasets of the independent test dataset were imbalanced, it was obtained from the real proteins and reflected the true distribution of pupylated sites and nonpupylated sites. A comparison of results with those of the existing methods, including the EPuL, PUL-PUP, PSoL, and SVM balance, is listed in Figure 1. The prediction results of PUL-PUP, PSoL, and SVM-balance are directly from EPuL [17]. we listed results of our method, PUL-PUP, PSoL, and SVM balance on the independent dataset in Appendix A.

The overall accuracy of the proposed method was the best among the five models. Especially, Sn reached 100% and Acc was 24% higher than that of EPuL. 

## 3. Materials and Methods

This study involved the following steps: (1) extracting amino acid composition (AAC); (2) calculating Pearson Correlation Coefficient and sorting the values; (3) selecting the top n amino acid pair composition; (4) combining with other features; (5) balancing the training datasets; and (6) verifying the model. The conceptual diagram of the pupylation site prediction model is shown in Figure 2. 

### 3.1. Formatting of Mathematical Components

The training and test datasets constructed by Tung [12] and lately used by Nan et al. [17], were also adopted in this study. The training dataset consisted of 183 experimentally validated pupylation sites and 2258 artificially generated nonannotated lysine sites from 162 proteins of Mycobacterium smegmatis (*M. smegmatis*), Mycobacterium tuberculosis (*M. tuberculosis*) and Escherichia coli (*E. coli*). The independent test dataset contained 29 experimentally verified lysine pupylation sites and 408 nonannotated lysine sites from 20 proteins. As the independent test dataset was highly unbalanced, it reflected the real effects of different methods [10,11,12,13,14,15,16,17]. Similar to the previous findings, the length of each protein peptide was 21 in the training and test datasets. The training and test datasets are available in the Appendix A.

### 3.2. Information from the Protein Peptide Sequence

#### 3.2.1. Amino Acid Composition (AAC)

AAC [27] is a widely used method for feature extraction, varied protein sequence analysis, and prediction. It was used to calculate the content of amino acids in the amino acid fragment. For a given peptide fragment containing 20 natural amino acids and pseudo amino acids "X," only 20 natural amino acids were used to construct the 20-dimensional feature vector of AAC for encoding because the frequency of pseudo amino acid "X" always approached zero or equaled zero. AAC can be defined as follows:(1)$$\mathrm{AAC}\;=\;\left[f_{1},f_{2},\ldots ,f_{20}\right]\;$$
where $\;f_{i}$ where $f_{i}\;$ represents the frequency of the occurrence of the *ith* amino acid in the 20 natural amino acids {A,C,D,E,F,G,H,I,K,L,M,N,P,Q,R,S,T,V,W,Y} and expressed as follows:(2)$$\;f_{i}=\;\frac{N_{i}}{N_{\mathrm{total}}}\;$$
where $N_{i}$ indicates the number of amino acid *i**th* in the peptide fragment, and $N_{total}$ indicates the total length of the peptide fragment. The length of the amino acid peptide was 20 in the study.

#### 3.2.2. Amino Acid Pair Composition

Amino acid pair composition (AAPC) [28] is generally used to obtain the correlation between one amino acid and other amino acids in a protein sequence. In this study, it was used to calculate the frequency of occurrence of amino acids in pupylation fragments. The AAPC feature vector is defined as follows: (3)$$\;\mathrm{AAPC}={\left[x_{1},x_{2},\ldots ,x_{i},\ldots ,x_{400}\right]}^{T}\;(0<i\leq 400)\;$$
where $\;x_{i}$ indicates the occurrence correlation of the *ith* amino acid pairs with other amino acid pairs (AA, AC, ..., YY).
(4)$$\;x_{i}\;=\;\frac{n_{ij}}{n_{total}}\;$$
where $n_{ij}$ indicates the number of amino acid pair *ijth* in the peptide fragment, and $n_{total}$ indicates the total number of amino acid pair. The length of the amino acid pair was 400 in the study.

#### 3.2.3. Adapted Normal Distribution bi-profile Bayes

ANBPB [19] is a combination of bi-profile Bayes (BPB) [29,30] and the standard normal distribution, which uses a probability vector to encode the peptide fragment.
(5)$$\;P_{j}={\left[p_{1},p_{2},\ldots ,p_{n},p_{n+1},\ldots ,p_{2n}\right]}^{T}\;$$
where $P_{j}(j=1,2,\ldots ,n)$ is the posterior probability of each amino acid in the *jth* position of the positive training dataset; $P_{j}(j=n+1,n+2,\ldots ,2n)$ is the posterior probability of each amino acid in the *jth* position of the negative training dataset. The occurrence frequency of each amino acid at each position is encoded as random variables $\;X_{ij\;}\left(i=1,2,\ldots ,21;\;j=1,2,\ldots ,21\right)$, which are independent and identically distributed in the binomial distribution  b=(m, p), in this study; *m = 183/2258* is the number of samples in the positive/negative dataset, p=1/21  is the probability of occurrence of each amino acid at each position. According to the Moivre–Laplace’s theorem, if *m* is large enough, $\frac{X_{ij\;}-\;mp}{\sqrt{mp\left(1-p\right)}}$ approximately obeys the standard normal distribution N(0,1). If $V_{j}$ is used to represent the standard deviation of the random variable $X_{ij\;}\left(i=1,2,\ldots ,21\right)$, the normalized variable of the random variable $X_{ij\;}$ can be defined as follows:(6)$$\;\;\;\;\;\;\;\;\;\;\;\;\;\;\;\;\;\;X_{ij}^{\prime }=\frac{X_{ij}-mp}{\sqrt{V_{j}}}.\;$$

Therefore, $p_{j\;}\left(i=1,2,\ldots n,n+1,\ldots ,2n\right)$ can be encoded by the adapted normal distribution as follows:(7)$$\;\;\;\;\;p_{j}=P\left(X\leq X_{ij}\right)=\varphi \left(X_{ij}^{\prime }\right)\;$$
where the formula for φ(x) is given by $\varphi \left(x\right)=\frac{1}{\sqrt{2\pi }}\overset{x}{\underset{-\infty }{\int }}e^{-\frac{t^{2}}{2}}dt$.

#### 3.2.4. TOP-n-gram

Top-n-gram [18], which includes the evolutionary information extracted from the frequency profiles, can be viewed as a novel profile-based building block of proteins. The multiple sequence alignments yielded by PSI-BLAST [31] are used to calculate the protein sequence frequency profiles, which are combined with the *n* most frequent amino acids in each amino acid frequency profile, and the frequency profiles are converted into Top-n-grams. The occurrence times of each TOP-n-gram are used to convert protein fragments into fixed feature vectors, and then the corresponding vectors are substituted into SVM. Several basic building blocks have been investigated as the words of "protein sequence language," including Ngrams [29,32], patterns [33], motifs [34], and binary profiles [18].

#### 3.2.5. Parallel Correlation Pseudo Amino Acid Composition

Parallel correlation pseudo amino acid composition (PC-PseAAC) was also considered to obtain more order information for a fragment [20]. PC-PseAAC is an approach incorporating the contiguous local sequence-order information and the global sequence-order information into the feature vector of the protein sequence. Given a protein sequence *P*, the PC-PseAAC feature vector of *P* is defined as follows [25]: (8)$$P\;=\;{\left[x_{1\;}x_{2}x_{3}\ldots x_{20}x_{21}\ldots x_{20+\lambda }\;\right]}^{T}\;$$
where
(9)$$\;\;\;\;x_{u}=\{\begin{array}{c} \frac{f_{u}}{\sum_{i=1}^{20}f_{i}+w\sum_{j=1}^{\lambda }\Theta_{j}}\left(1\leq u\leq 20\right)\\ \frac{{w\Theta }_{u-20}}{\sum_{i=1}^{20}f_{i}+w\sum_{j=1}^{\lambda }\Theta_{j}}\left(20+1\leq u\leq 20+\lambda \right) \end{array}\;$$
where *fi (i = 1,2, …,20)* is the normalized occurrence frequency of the 20 amino acids in the protein sequence *P*; the parameter *λ* is an integer, representing the highest counted rank (or tier) of the correlation along a protein sequence; *w* is the weight factor ranging from 0 to 1; and *θj (j = 1, 2,*
*⋯,*
*λ)* is called the *j-tier* correlation factor reflecting the sequence-order correlation between all the *jth* most contiguous residues along a protein chain and defined as follows:(10)$$\;\Theta_{\lambda }=\frac{1}{L-\lambda }\overset{L-\lambda }{\underset{i=1}{\sum }}\Theta (R_{i},R_{i+\lambda })\left(0<\lambda <1\right)\;$$
where the correlation function is given by:(11)$$\;\Theta (R_{i},R_{j})=\frac{1}{3}\left\{{\left[H_{1}\left(R_{j}\right)-H_{1}\left(R_{i}\right)\right]}^{2}+{\left[H_{2}\left(R_{j}\right)-H_{2}\left(R_{i}\right)\right]}^{2}+{\left[M\left(R_{j}\right)-M\left(R_{i}\right)\right]}^{2}\right\}\;$$
where $\;H_{1}\left(R_{i}\right),\;H_{2}\left(R_{i}\right),\;and\;M\left(R_{i}\right)$ are the hydrophobicity value, hydrophilicity value, and side-chain mass of the amino acid $R_{i}$, respectively. Before substituting the values of hydrophobicity, hydrophilicity, and side-chain mass into [18], they should all be subjected to a standard conversion as described by the following equation [25]:(12)$$\;\;\;\;\;H_{1}\left(i\right)=\frac{H_{1}^{0}\left(i\right)-\sum_{i=1}^{20}\frac{H_{1}^{0}\left(i\right)}{20}}{\sqrt{\frac{\sum_{i=1}^{20}{\left[H_{1}^{0}\left(i\right)-\sum_{i=1}^{20}\frac{H_{1}^{0}\left(i\right)}{20}\right]}^{2}}{20}}}\;$$
(13)$$\;H_{2}\left(i\right)=\frac{H_{2}^{0}\left(i\right)-\sum_{i=1}^{20}\frac{H_{2}^{0}\left(i\right)}{20}}{\sqrt{\frac{\sum_{i=1}^{20}{\left[H_{2}^{0}\left(i\right)-\sum_{i=1}^{20}\frac{H_{2}^{0}\left(i\right)}{20}\right]}^{2}}{20}}}\;$$
(14)$$\;M\left(i\right)=\frac{M^{0}\left(i\right)-\sum_{i=1}^{20}\frac{M^{0}\left(i\right)}{20}}{\sqrt{\frac{\sum_{i=1}^{20}{\left[M^{0}\left(i\right)-\sum_{i=1}^{20}\frac{M^{0}\left(i\right)}{20}\right]}^{2\;}}{20}}}\;$$
where $H_{1}^{0}\left(i\right)$ is the original hydrophobicity value of the *ith* amino acid; $H_{2}^{0}\left(i\right)$ is the corresponding original hydrophilicity value; and $M^{0}\left(i\right)\;is\;$ the mass of the *ith* amino acid side chains.

### 3.3. Pearson Correlation Coefficient

As a measure of the variability among variables, the correlation coefficient can indicate a certain correlation between two variables at the same time. The Pearson correlation coefficient [23,24] is used to judge whether two datasets are on a line and measure the linear relationship according to distance. For the Pearson correlation coefficient, if a positive linear correlation exists between two variables, their metric value approaches 1. Similarly, if the two variables have a negative linear correlation, their metric value approaches –1. Let *(X, Y)* be a two-dimensional random variable vector, and the Pearson correlation coefficient is defined as follows:(15)$$\;\rho =\frac{\mathrm{cov}\;\left(\;X,\;Y\;\right)}{\sqrt{\mathrm{var}\left(X\right)}\;\sqrt{\mathrm{var}\left(Y\right)}}\;$$

Let ($X_{i},Y_{i}$) (i = 1, 2 … n ) be a random sample of  (X, Y), and another form of the Pearson correlation coefficient can be expressed as follows:(16)$$\;\rho \;=\frac{\sum_{i=1}^{n}\left(X_{i}-\bar{X}\right)\left(Y_{i}-\bar{Y}\right)}{{\left[\sum_{i=1}^{n}{\left(X_{i}-\bar{X}\right)}^{2}\;\sum_{i=1}^{n}{\left(Y_{i}-\bar{Y}\right)}^{2}\right]}^{1/2}}\;$$
where $\bar{X}\;$ is the mean of $X_{i}$, and $\bar{Y\;}$ is the mean of $Y_{i},(i$
*= 1, 2 …*
n).

In this study, AAC was used for feature extraction on the training dataset, and the values of Pearson Correlation Coefficients were sorted in a descending order. The positive and negative training datasets were extracted in AAPC. The feature selection was performed on the 400-dimensional feature vector, and the best results were obtained when the first AAPC (320) features were selected. 

### 3.4. Processing of Positive and Negative Datasets

#### 3.4.1. KPCA Oversampling Technique

The KPCA oversampling technique [21] is based on the K-means clustering algorithm and principal component analysis (PCA) [35,36]. First, the initial clustering center *K* was formed by randomly selecting class samples from the positive training dataset.
(17) K(InitialCenter(j)),j=1,2,…n 

Second, the Euclidean distance was used to divide the positive training samples into *k* clusters, defined as follows:(18)$$\;I_{j}=||K\left(j\right)-K{\left(\mathrm{InitialCenter}\left(n\right)\right)||}^{2}\;$$
where K(j) represents *j* original positive training samples, and $I_{j}$ is the distance between *j* original positive training samples and *n* clusters.

Finally, the average values of all peptide fragments of the original positive training sample were calculated in each cluster and then clustering centers were adjusted. The initial clustering center *K* was not picked twice, and the process was performed iteratively. Next, each cluster of positive training samples $\;\;K={\left[k_{1},\;k_{2},\ldots ,k_{m1}\right]}^{T}$ was normalized by $K^{\prime }=\frac{k_{i}-\mu }{\sigma }\;$ to be $K^{\prime }\;$*=*$\;{\left[k_{1}^{\prime },k_{2}^{\prime },\ldots ,\;k_{m1}^{\prime }\right]}^{T}$. Then, the covariance matrix R =$\;{\left(r_{ij}\right)}_{m*n}\;\;of\;\;K^{\prime }$ was obtained and the following equations were used to add the synthesized positive samples to the positive training dataset.
(19)$$\;\;K^{\prime }={\left[k_{1}^{\prime },k_{2}^{\prime },\ldots ,\;k_{m1}^{\prime }\right]}^{T}\;$$
(20)       rij=(∑i=1mXkj∗Xkj)(∑k=1m(Xki)2∑k=1m(Xkj)2)12   
(21)$$\;Y={\left[y_{1},\;y_{2\;},\ldots ,y_{m1}\right]}^{T}\;$$
(22)$$\;y_{i}=\left(y_{i1},y_{i2},\ldots y_{i\mathrm{Dim}}\;\right),i=1,2,\ldots ,\;\mathrm{Dim}\;$$
(23)$$y_{ij}=k_{i}^{\prime }*A\left(:,j\right),i=1,2,\ldots ,\;m1,\;j=1,2,\;\ldots ,\;\mathrm{Dim}\;$$
where *m1* is the number of positive samples in the *nth* cluster, Dim is the number of features in the positive samples, and *A* is the eigenvalue matrix of the covariance matrix arranged in a descending order. In this study, the value of *K* was set to 3 and KPCA was used on the positive samples. Finally, the experiment of clustering was repeated 4 times and their results were added to 183 positive training datasets as the new positive training dataset, including 377 samples. The number of the new positive training samples and that of the new negative training samples obtained from 365 samples using the FUS approach 1:1.

#### 3.4.2. Fuzzy Undersampling Method

The FUS method [22] uses the fuzzy membership function to extract the information hidden in the training samples. In this study, the FUS approach was used to reduce the number of negative training samples to keep the positive and negative datasets in balance.

First, the mean and standard deviation of each feature of the positive and negative training samples were calculated as follows:(24)$$\;C_{\mathrm{pos}}^{j}=\frac{\sum_{i=1}^{\mathrm{PosNum}}\mathrm{Pos}\left(i,j\;\right)}{\mathrm{PosNum}},j=1,2,\ldots ,\mathrm{Dim}\;$$
(25)$$\;C_{\mathrm{Neg}}^{j}=\frac{\sum_{i=1}^{\mathrm{NegNum}}\mathrm{Neg}\left(i,j\;\right)}{\mathrm{NegNum}},j=1,2,\ldots ,\mathrm{Dim}\;$$
(26)$$\;\sigma_{\mathrm{Pos}}^{j}=\sqrt{\frac{1}{\mathrm{PosNum}}\overset{\mathrm{PosNum}}{\underset{i=1}{\sum }}{\left(\mathrm{Pos}\left(i,j\right)-C_{\mathrm{Pos}}^{j}\right)}^{2}\;},j=1,2,\ldots ,\mathrm{Dim}\;$$
(27)$$\;\;\sigma_{\mathrm{Neg}}^{j}=\sqrt{\frac{1}{\mathrm{NegNum}}\overset{\mathrm{NegNum}}{\underset{i=1}{\sum }}{\left(\mathrm{Neg}\left(i,j\right)-C_{\mathrm{Neg}}^{j}\right)}^{2}\;},j=1,2,\ldots ,\mathrm{Dim}\;$$
where PosNum is the number of positive training samples, NegNum is the number of negative training samples, Pos(i,j ) is the value of the *jth* feature of the *ith* positive training sample, and  Neg(i,j ) is the value of the *jth* feature of the *ith* negative training sample. *Dim* is the number of training samples features.

Second, the mean and standard deviation of the positive and negative samples were used to establish a membership function, defined as follows:(28)$$\;\;u_{\mathrm{Pos}}^{j}\left(i\right)=\mathrm{GaussMF}\left(\mathrm{Data}\left(i,j\right);C_{\mathrm{Pos}}^{j},\sigma_{\mathrm{Pos}}^{j}\right)\;$$
(29)$$\;u_{\mathrm{Neg}}^{j}\left(i\right)=\mathrm{GaussMF}\left(\mathrm{Data}\left(i,j\right);C_{\mathrm{Neg}}^{j},\sigma_{\mathrm{Neg}}^{j}\right)\;$$
(30)$$\;\mathrm{GaussMF}\left(x;C,\sigma \right)=\exp\left(-\frac{1}{2}{\left(\frac{x-C}{\sigma }\right)}^{2}\right)\;$$
where Data(i,j) is the jth eigenvalue of the ith sample among the training samples (i=1,2,…,PosNum,PosNum+1,…,PosNum+NegNum, j=1,2,…,Dim).
(31)$$\;\mathrm{Fval}\left(i,j\right)=u_{\mathrm{Pos}}^{j}\left(i\right)+\left(1-u_{\mathrm{Neg}}^{j}\left(i\right)\right).\;$$

Finally, all the positive training samples were saved and the negative training samples with high scores were removed. The score function is defined as follows:(32)$$\;\mathrm{Score}\left(i\right)=\overset{\mathrm{Dim}}{\underset{j=1}{\sum }}\mathrm{Fval}\left(i,j\right),\left(i=\;\mathrm{PosNum}+1,\ldots ,\mathrm{PosNum}+\mathrm{NegNum}\right).\;$$

Then, FUS was applied on the negative training samples, and the number of negative training samples was reduced from 2258 to 365, which could decrease the inaccuracy caused by imbalanced datasets.

### 3.5. SVMs Implementation and Parameter Selectiont

SVM is a supervised learning method, which applies statistical theory to complete classification and regression in the area of bioinformatics [15,37,38,39,40,41,42,43]. In this study, the LibSVM package [43,44] was used to establish and evaluate the performance of the prediction model. The kernel function used the radial basis kernel function (RBF) $K\left(S_{i},S_{j}\right)=e^{\left(-\gamma \|S_{i}-S_{j}{\|}^{2}\right)}$, and a grid search strategy based on 10 times of tenfold cross-validation and jackknife was employed to find the optimal parameters. The optimal parameters of tenfold cross-validation and jackknife are *C* = 0.70711 and *γ*= 1.4142, respectively.

Four measurements, including sensitivity (Sn), specificity (Sp), accuracy (Acc), and Matthews correlation coefficient (MCC), were used to evaluate the performance of the proposed predictor [44], which are shown as follows:(33)$$\mathrm{Sn}=\frac{\mathrm{TP}}{\mathrm{TP}\;+\;\mathrm{FN}\;}\;\;$$
(34)$$\;\mathrm{Sp}=\frac{\mathrm{TN}}{\mathrm{TN}\;+\;\mathrm{FP}}\;$$
(35)$$\;\mathrm{Acc}=\frac{\mathrm{TP}\;+\;\mathrm{TN}}{\mathrm{TP}\;+\;\mathrm{TN}\;+\;\mathrm{FP}\;+\;\mathrm{FN}}\;$$
(36)$$\;\mathrm{MCC}=\frac{\mathrm{TP}\;\times \;\mathrm{TN}\;-\;\mathrm{FN}\;\times \;\mathrm{FP}}{\sqrt{\left(\mathrm{TP}\;+\;\mathrm{FN}\right)\;\times \;\left(\mathrm{TP}\;+\;\mathrm{FP}\right)\;\times \;\left(\mathrm{TN}\;+\;\mathrm{FP}\right)\;\times \;\left(\mathrm{TN}\;+\;\mathrm{FN}\right)}}\;$$
where TP, FP, TN, and FN are the number of true positives, false positives, true negatives, and false negatives, respectively.

## 4. Conclusions

In this study, the Pearson correlation coefficient was used to evaluate the relevance between any two amino acids, which were employed to obtain an optimal combination of amino acid pairs. Then, the AAPC (320) information was combined with TOP-n-gram, ANBPB, and PC-PseAAC to construct multiple feature vectors. Moreover, KPAC and FUS were used to solve the imbalance between positive and negative datasets. Finally, tenfold cross-validation, jackknife test, and an independent test were used to verify the proposed model. The prediction results showed that the proposed method was more accurate than the previous methods in predicting the pupylation sites. However, this study only considered the content of amino acids and neglected the location of amino acids in pupylated proteins. Amino acid environment such as surface areas and buried residues should be considered in future studies to explore the effects of amino acids at different positions. It should be pointed out user-friendly and publicly accessible web-servers will significantly enhance their impacts, we shall make efforts in our future work to provide a web-server for the prediction method presented in this paper. The MATLAB (matrix laboratory) package of our prediction method is available as Appendix A.

## Figures and Tables

**Figure 1 molecules-23-03097-f001:**
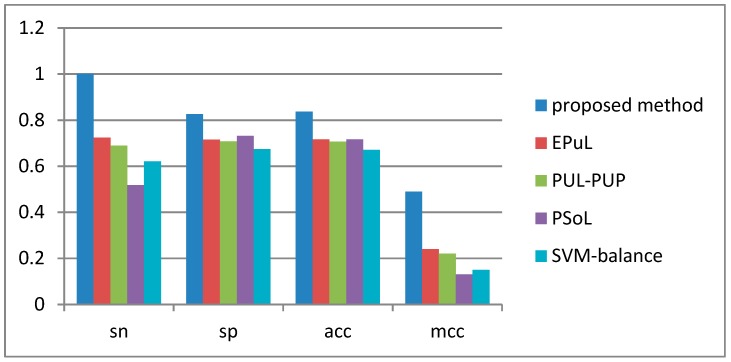
Comparison of performance between our model, EPuL, PUL-PUP, PSoL, and SVM balance on an independent test dataset.

**Figure 2 molecules-23-03097-f002:**
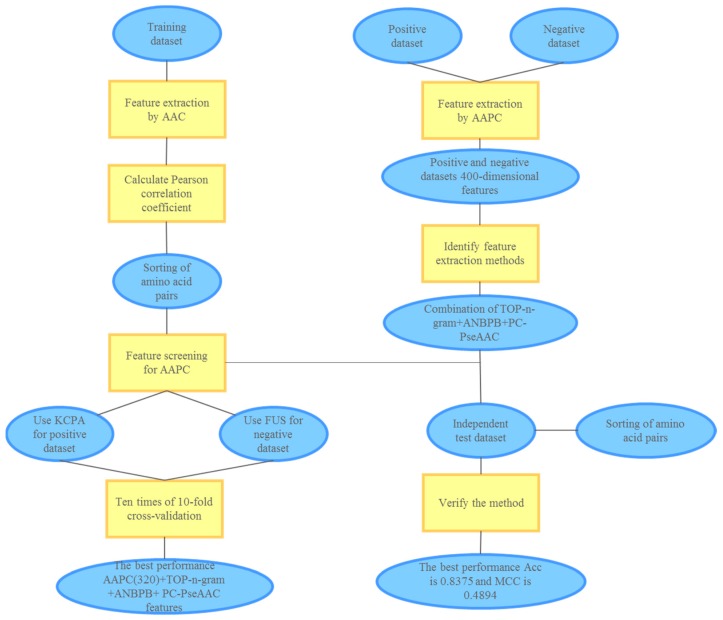
Conceptual diagram of the pupylation site prediction model.

**Table 1 molecules-23-03097-t001:** Comparison of original imbalance dataset and balanced dataset.

Method	Sn (%)	Sp (%)	Acc (%)	MCC
Without resampling	0	100	92.50	NaN
KPCA oversampling	25.54	99.87	92.67	0.33
KPCA oversampling and FUS undersampling	81.42	76.44	78.10	0.55

**Table 2 molecules-23-03097-t002:** Predictive performance of TOP-n-gram, adapted normal distribution bi-profile Bayes (ANBPB), and parallel correlation pseudo amino acid composition (PC-PseAAC) using the jackknife test.

Feature	Sn (%)	Sp (%)	Acc (%)	MCC
AAPC(320)	81.42	76.44	78.1	0.55
AAPC(320) + TOP-n-gram	80.33	100	93.43	0.86
AAPC(320) + ANBPB	64.48	100	88.14	0.74
AAPC(320) + PC-PseAAC	84.15	72.60	76.46	0.54
AAPC(320) + TOP-n-gram + ANBPB	70.49	98.36	89.05	0.75
AAPC(320) + TOP-n-gram + PC-PseAAC	87.43	75.34	79.38	0.59
AAPC(320) + ANBPB + PC-PseAAC	75.41	100	91.79	0.82
AAPC(320) + TOP-n-gram + ANBPB + PC-PseAAC	94.54	100	98.18	0.96

**Table 3 molecules-23-03097-t003:** Comparison of the proposed method with EPuL, PUL-PUP, PSoL, and support vector machine (SVM) balance on 10-fold cross-validation test.

Method	Sn (%)	Sp (%)	Acc (%)	MCC	AUC
Proposed method	90.53	99.8	95.09	0.91	0.96
* EPuL	84.21	95.45	90.24	0.81	0.93
* PUL-PUP	82.24	91.57	88.92	0.74	0.92
* PSoL	67.5	73.6	70.55	0.42	0.8
* SVM balance	76.71	63.65	69.88	0.4	0.77

* The results of the comparison are from EPuL [17].

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
