# Peer review of "Recognition of Protein Pupylation Sites by Adopting Resampling Approach"

_molecules, 2018, doi:10.3390/molecules23123097_

Round 1
Reviewer 1 Report
The authors present an interesting alternative approach to predict the pupylation. The methods utilized are technically well performed however there may be some shortcomings.
Lines 36-38: The authors allude the ubiquitination in eukaryotic DNA repair. As such, it
would be worth expanding on the role of ubiquitination in eukaryotic DNA repair, particularly recent advances on its regulatory role for careful series of functions that gives rise to a therapeutic solution in cancer chemotherapy, indicating the significance of ubiquitination studies.
Lines 36-41: The authors have done little effort of reviewing the artciles (only 4 articles citation in introduction) on the ubiquitination in eukaryotes and pupylation in prokaryotes. There are thousands of literatures on ubiquitination that authors could study and cite a selection of those.
The authors can consider adding a statement like below that gives credit to the knowledge obtained and that how it is relevant to ubiquitination prediction:
“Since experimental methods are laborious, it is essential to improve the current computational methodologies to provide direction for further research on intriguing research questions. An example of this is the works on understanding the stability of ERCC1 DNA repair protein, a biomarker of several advanced cancer diseases. This protein functions in multiple DNA repair pathways and certain mutations in this protein and its partner had drastic consequences for the protein complex stability (1-8). Functional and structural studies have determined the significance of the complex integrity and ubiquitination/deubiquitination events for controlling the function of the protein during DNA damage response (9-12). These findings persuade new therapeutic solution in cancer chemotherapy. Thus, it is important to devise scientist with accurate computational methodologies to ultimately predict the probability of different regulatory pathways that control protein function and disease severities.”
1. Friedberg E. C., Walker G. C., Siede W. (1995) DNA Repair and Mutagenesis, pp. 135–190, American Society of Microbiology Press, Washington, D. C.
2. Bauman JE, Austin MC, Schmidt R, Kurland BF, Vaezi A, Hayes DN, Mendez E, Parvathaneni U, Chai X, Sampath S, Martins RG. (2013) ERCC1 is a prognostic biomarker in locally advanced head and neck cancer: results from a randomised, phase II trial. Br J Cancer 109(8):2096-105.
3. Lee SM, Falzon M, Blackhall F, Spicer J, Nicolson M, Chaudhuri A, Middleton G, Ahmed S, Hicks J, Crosse B, Napier M, Singer JM, Ferry D, Lewanski C, Forster M, Rolls SA, Capitanio A, Rudd R, Iles N, Ngai Y, Gandy M, Lillywhite R, Hackshaw A1 (2017) Randomized Prospective Biomarker Trial of ERCC1 for Comparing Platinum and Nonplatinum Therapy in Advanced Non–Small-Cell Lung Cancer: ERCC1 Trial (ET). Journal of Clinical Oncology 35, no. 4 402-411.
4. Sijbers A. M., de Laat W. L., Ariza R. R., Biggerstaff M., Wei Y.-F., Moggs J. G., Carter K. C., Shell B. K., Evans E., de Jong M. C., Rademakers S., de Rooij J., Jaspers N. G. J., Hoeijmakers J. H. J., Wood R. D. (1996) Xeroderma pigmentosum group F caused by a defect in a structure-specific DNA repair endonuclease. Cell 86, 811–822
5. Al-Minawi A.Z., Lee Y.-F., Håkansson D., Johansson F., Lundin C., Saleh-Gohari N., Schultz N., Jenssen D., Bryant H. E., Meuth M., Hinz J. M., Helleday T. (2009) The ERCC1/XPF endonuclease is required for completion of homologous recombination at DNA replication forks stalled by inter-strand cross-links. Nucleic Acids Res. 37, 6400–6413
6. Ahmad A., Robinson A. R., Duensing A., van Drunen E., Beverloo H. B., Weisberg D. B.,Hasty P., Hoeijmakers J. H. J., Niedernhofer L. J. (2008) ERCC1-XPF endonuclease facilitates DNA double-strand break repair. Mol. Cell Biol. 28, 5082–5092
7. Al-Minawi A. Z., Saleh-Gohari N., Helleday T. (2008) The ERCC1/XPF endonuclease is required for efficient single-strand annealing and gene conversion in mammalian cells. Nucleic Acids Res. 36, 1–9
8. Faridounnia, M., Wienk, H., Kovačič, L., Folkers, G. E., Jaspers, N. G., Kaptein, R., Hoeijmakers, J. H., and Boelens, R. (2015) The Cerebro-Oculo-Facio-Skeletal (COFS) Syndrome point mutation F231L in the ERCC1 DNA repair protein causes dissociation of the ERCC1-XPF complex. Journal of Biological Chemistry 290, 20541-20555
9. van Cuijk, L., Vermeulen, W., and Marteijn, J. A. (2014) Ubiquitin at work: the ubiquitous regulation of the damage recognition step of NER. Experimental cell research 329, 101-109
10. Zhang, L., and Gong, F. (2016) The emerging role of deubiquitination in nucleotide excision repair. DNA Repair (Amst) 44, 118-122
11. van Cuijk, L., van Belle, G. J., Turkyilmaz, Y., Poulsen, S. L., Janssens, R. C., Theil, A. F., Sabatella, M., Lans, H., Mailand, N., Houtsmuller, A. B., Vermeulen, W., and Marteijn, J. A. (2015) SUMO and ubiquitin-dependent XPC exchange drives nucleotide excision repair. Nat Commun 6, 7499
12. Perez-Oliva, A. B., Lachaud, C., Szyniarowski, P., Munoz, I., Macartney, T., Hickson, I., Rouse, J., and Alessi, D. R. (2015) USP45 deubiquitylase controls ERCC1-XPF endonuclease-mediated DNA damage responses. The EMBO journal 34, 326-343
Line 42 and 43: The citations for these two statements are missing. Please cite the relevant works, particularly line 42 needs citations as several proteomics efforts are mentioned.
Line 316: It is recommended to compile the prediction codes in generating a website and provide the link to the website in the paper, so that the biologists can refer to and take advantage of the method. Also, the purpose of the research can be served in this way (by providing the audience with means to predict the pupylation sites on the protein of interest). In case making a bioinformatic tool is a future plan, authors could make the current methodology in the form of codes, scripts and command line.
Line 318: More perspective should be added on what algorithm will be used and if that is not yet determined it is recommended to take amino acid environment into account and simply recommend the use of current algorithm with bioinformatic tools for predicting surface areas and buried residues, which helps with determination of exposed amino acid residues for further analysis. It is highly recommended to reword lines 317-319.
The font size in reference 42 is different from the rest of the text.
Author Response
Response to Reviewer 1 Comments
Point 1: Lines 36-38: The authors allude the ubiquitination in eukaryotic DNA repair. As such, it would be worth expanding on the role of ubiquitination in eukaryotic DNA repair, particularly recent advances on its regulatory role for careful series of functions that gives rise to a therapeutic solution in cancer chemotherapy, indicating the significance of ubiquitination studies.
Response 1: Thanks for your suggestion. We added the following description:
“Its proteasome-independent functions help in the regulation of DNA repair mechanisms, particularly recent advances on its regulatory role for careful series of functions that gives rise to a therapeutic solution in cancer chemotherapy [3-5].”
Point 2: Lines 36-41: The authors have done little effort of reviewing the articles (only 4 articles citation in introduction) on the ubiquitination in eukaryotes and pupylation in prokaryotes. There are thousands of literatures on ubiquitination that authors could study and cite a selection of those.
Response 2: Thank you for your comments. We have added all of the references and updated the references in the revised manuscript.
Point 3: The authors can consider adding a statement like below that gives credit to the knowledge obtained and that how it is relevant to ubiquitination prediction:
“Since experimental methods are laborious, it is essential to improve the current computational methodologies to provide direction for further research on intriguing research questions. An example of this is the works on understanding the stability of ERCC1 DNA repair protein, a biomarker of several advanced cancer diseases. This protein functions in multiple DNA repair pathways and certain mutations in this protein and its partner had drastic consequences for the protein complex stability (1-8). Functional and structural studies have determined the significance of the complex integrity and ubiquitination/deubiquitination events for controlling the function of the protein during DNA damage response (9-12). These findings persuade new therapeutic solution in cancer chemotherapy. Thus, it is important to devise scientist with accurate computational methodologies to ultimately predict the probability of different regulatory pathways that control protein function and disease severities.”
Response 3: We have added the above statement in Line 40.
Point 4: This Line 42 and 43: The citations for these two statements are missing. Please cite the relevant works, particularly line 42 needs citations as several proteomics efforts are mentioned.
Response 4: Thanks for your advice. We have carefully revised the manuscript. References [11-17] were added at line 44 to introduce the work of related proteomics, which is helpful for readers to understand. And add the transition sentence at line 52:” In order to better recognize the occurrence site of advanced cancer, accurate identification of the proteinization site is the first step to understand the proteinization mechanism.”
Point 5: Authors Line 316: It is recommended to compile the prediction codes in generating a website and provide the link to the website in the paper, so that the biologists can refer to and take advantage of the method. Also, the purpose of the research can be served in this way (by providing the audience with means to predict the pupylation sites on the protein of interest). In case making a bioinformatic tool is a future plan, authors could make the current methodology in the form of codes, scripts and command line.
Response 5: It should be pointed out user-friendly and publicly accessible web-servers will significantly enhance their impacts, we shall make efforts in our future work to provide a web-server for the prediction method presented in this paper. And the prediction code was upload as supplementary.
Point 6: Line 318: More perspective should be added on what algorithm will be used and if that is not yet determined it is recommended to take amino acid environment into account and simply recommend the use of current algorithm with bioinformatic tools for predicting surface areas and buried residues, which helps with determination of exposed amino acid residues for further analysis. It is highly recommended to reword lines 317-319.
Response 6: Thanks for your valuable suggestions. We have reworded lines 317-319 as follows.
“Amino acid environment such as surface areas and buried residuesshould be considered in future studies to explore the effects of amino acids at different positions”
Point 7: Authors mentioned that the font size in reference 42 is different from the rest of the text.
Response 7: Sorry for the mistake. We have carefully revised the manuscript.
Reviewer 2 Report
The paper by Li and colleagues present a tool to predict pupylation sites. They present a solid study, and the manuscript is well written. The results are significantly better compared to other methods.
Major concerns:
- Please make the software available, through GitHub or similar, and do not forget to specify the license. Provide links and license information in the manuscript
- Please add a supplemental tables detailing the reduction of the true negatives, and, more importantly, the prediction results for all methods.
- Are all the comparisons made using re-implementations of the original algorithms? This is somewhat unclear, so please report more explicitly which algorithms were reimplemented and in which cases the original tools were used. Please report how similar the re-implementation performed to the original algorithm. If this comparison cannot be done (missing tool, not installable, etc.), please report this explicitly.
Minor concerns:
Unclear, row 84: "Finally, the AAPC was selected according to the order."
Row 113: "on the web server" please write out the "name" of the web server in addition to the reference.
Row 156: please write a more comprehensive figure legend.
Author Response
Response to Reviewer 2 Comments
Major concerns:
Point 1: Please make the software available, through GitHub or similar, and do not forget to specify the license. Provide links and license information in the manuscript.
Response 1: we have revised the paper.
Point 2: Please add a supplemental tables detailing the reduction of the true negatives, and, more importantly, the prediction results for all methods.
Response 2: Many Thanks. We have upload the reduction of the true negatives as ?. And the prediction results on unbalanced dataset and balanced datasets were listed in Table 1.
Table 1. Comparison of AAPC(320) and AAPC(320) + KPCA + FUS using the jackknife test.
Method | Sn (%) | Sp (%) | Acc(%) | MCC |
AAPC(320) | 0 | 1 | 92.50 | NaN |
AAPC(320) + KPCA | 94.54 | 99.87 | 92.67 | 0.33 |
AAPC(320) + KPCA + FUS | 81.42 | 76.44 | 78.10 | 0.55 |
Point 3: Authors mentioned that Are all the comparisons made using re-implementations of the original algorithms? This is somewhat unclear, so please report more explicitly which algorithms were reimplemented and in which cases the original tools were used. Please report how similar the re-implementation performed to the original algorithm. If this comparison cannot be done (missing tool, not installable, etc.), please report this explicitly.
Response 3: Thank you for your comments. We added the annotation under Table 3 and the above of Figure 1.
Minor revisions:
Point 1: Unclear, row 84: "Finally, the AAPC was selected according to the order."
Response 1: We agree with your suggestion, so we have rewritten this part as follow: Finally, the features of AAPC(400)was sorted according to the order, and the AAPC was selected.
Point 2: Row 113: "on the web server" please write out the "name" of the web server in addition to the reference.
Response 2: Many thanks. We rewrite this part as follows:
“A variety of feature extraction methods can be found on the Web server [23] (http://bioinformatics.hitsz.edu).”
Point 3: Row 156: please write a more comprehensive figure legend.
Response 3: Thanks for your comments. We change Figure 1 as follows:
Figure 1. Comparison of performance between our model, EPuL, PUL-PUP, PSoL and SVM balance on an independent test dataset.

Reviewer 3 Report
The authors present an algorithm for predicting protein modification sites (prokaryotic pupylation) based on a machine-learning approach for identifying sequence features that correlate with known pupylation sites. The approach is compared to existing prediction algorithms and tested on a new dataset.
The study's approach appears appropriate and rigorous. However, the presentation was very difficult to follow; I didn't understand the model at all until reading the methods section and equations in detail. I would suggest that the authors more clearly explain their model in the abstract, introduction and results sections of the text, so that the reader can more easily follow the results and understand their significance.
Author Response
Response to Reviewer 3 Comments
Point 1: The study's approach appears appropriate and rigorous. However, the presentation was very difficult to follow; I didn't understand the model at all until reading the methods section and equations in detail. I would suggest that the authors more clearly explain their model in the abstract, introduction and results sections of the text, so that the reader can more easily follow the results and understand their significance.
Response 1: Thanks for your suggestions. We have explained our model in the abstract, introduction and results sections, respectively.
“Abstract: With the in-depth study of posttranslational modification sites, protein ubiquitination has become the key problem to study the molecular mechanism of posttranslational modification. Pupylation is a widely used process in which a prokaryotic ubiquitin-like protein (Pup) is attached to a substrate through a series of biochemical reactions. However, the experimental methods of identifying pupylation sites is often time-consuming and laborious. This study aims to propose an improved approach for predicting pupylation sites. Firstly, the Pearson correlation coefficient was used to reflect the correlation among different amino acid pairs calculated by the frequency of each amino acid. Then according to a descending ranked order, the multiple types of features were filtered separately by values of Pearson correlation coefficient. Thirdly, to get a qualified balanced dataset, the K-means principal component analysis (KPCA) oversampling technique was employed to synthesize new positive samples and Fuzzy undersampling method was employed to reduce the number of negative samples. Finally, the performance of our method was verified by means of jackknife and 10-fold cross-validation test. The average results of 10-fold cross-validation showed that the sensitivity (Sn) was 90.53%, specificity (Sp) was 99.8%, accuracy (Acc) was 95.09%, and Matthews Correlation Coefficient (MCC) was 0.91. Moreover, an independent test dataset was used to further measure its performance, and the results showed that the Acc was 83.75%, MCC was 0.49, which was superior to previous predictors. The proposed method showed better performance and stability, proving as an effective way to predict pupylation sites.
The last part of Introduction is revised as follows:
“In this study, the Pearson correlation coefficient was used to evaluate the relevance among amino acid pairs based on amino acid composition information. Then, according to the descending order, the amino acid pair features were added by a step of 20 to get the best prediction results by means of jackknife test. Then the selected 320 amino acid pairs were combined with TOP-n-gram [18], Adapted normal distribution bi-profile Bayes (ANBPB) [19], and Parallel correlation pseudo amino acid composition (PC-PseAAC) [20] to construct a multiple feature vector for the query protein peptide sequence. Because of the imbalance between the number of positive and negative samples (183:2258), the K-means principal component analysis (KPCA) oversampling technique, firstly proposed by Jia and Zuo [21], was applied on the positive training dataset for oversampling, and the synthetic samples were added to the original positive training dataset as a new positive training dataset. The fuzzy undersampling (FUS) method [21, 22] was applied to the negative dataset to reduce noise negative samples, and the selected negative samples were used as a new negative training dataset. At last, the 377 pupylated sites (positive samples) and 365 non-pupylated lysine sites (negative samples) were used to train and test our model. Moreover, the performance of our method was verified by means of jackknife, 10-fold cross-validation test and an independent dataset. When compared with other existing predictors, all of the results showed better performance and stability, proving as an effective way to predict pupylation sites.”
Round 2
Reviewer 2 Report
The author's, unfortunately, did not properly address the concerns raised in the first review. I am concerned that they report that the addressed the concern when they did not do this.
From the rebuttal letter:
"Point 1: Please make the software available, through GitHub or similar, and do not forget to specify the license. Provide links and license information in the manuscript.
Response 1: we have revised the paper. "
The code is still not available in a public repository such as GitHub. Make the code available including sufficient information on how to install and test it. Please specify the license.
From the rebuttal letter:
"Point 2: Please add a supplemental tables detailing the reduction of the true negatives, and, more importantly, the prediction results for all methods."
There is now a supplemental table, Table S1, which is good but is not sufficient. I am looking for a table that lists each site in each protein if this is part of the training or test data set, and the prediction results for each method. This table should have 183+2258 rows for the training dataset and 29+408 rows for the test dataset. It should minimally have the following columns: protein, site, verified, test/training, proposed method result/score, EPuL result/score, PUL-PUP result/score, PSoL result/score, SVM-balance result/score.
Author Response
Point 1: The code is still not available in a public repository such as GitHub. Make the code available including sufficient information on how to install and test it. Please specify the license.
Response 1: I am sorry for the carelessness. We have attached the Predictor as the supplementary.
Point 2: Please add a supplemental tables detailing the reduction of the true negatives, and, more importantly, the prediction results for all methods.
There is now a supplemental table, Table S1, which is good but is not sufficient. I am looking for a table that lists each site in each protein if this is part of the training or test data set, and the prediction results for each method. This table should have 183+2258 rows for the training dataset and 29+408 rows for the test dataset. It should minimally have the following columns: protein, site, verified, test/training, proposed method result/score, EPuL result/score, PUL-PUP result/score, PSoL result/score, SVM-balance result/score.
Response 2: Thanks for your suggestion. We have attached the training and test datasets as Supplementary. For the EPuL web-server is unavailable, we listed the detailed 10-fold cross-validation results and the independent test of our method, PUL-PUP, PSoL and SVM balance in Supplementary Table S2-S9.